# Pharmacological Characterisation of *Pseudocerastes* and *Eristicophis* Viper Venoms Reveal Anticancer (Melanoma) Properties and a Potentially Novel Mode of Fibrinogenolysis

**DOI:** 10.3390/ijms22136896

**Published:** 2021-06-27

**Authors:** Bianca op den Brouw, Parviz Ghezellou, Nicholas R. Casewell, Syed Abid Ali, Behzad Fathinia, Bryan G. Fry, Mettine H.A. Bos, Maria P. Ikonomopoulou

**Affiliations:** 1Venom Evolution Lab, School of Biological Sciences, The University of Queensland, St. Lucia, QLD 4072, Australia; bgfry@uq.edu.au; 2Medicinal Plants and Drugs Research Institute, Shahid Beheshti University, Tehran 1983969411, Iran; parviz.ghezellou@anorg.chemie.uni-giessen.de; 3Institute of Inorganic and Analytical Chemistry, Justus Liebig University Giessen, 35392 Giessen, Germany; 4Centre for Snakebite Research & Interventions, Liverpool School of Tropical Medicine, Liverpool L3 5QA, UK; Nicholas.Casewell@lstmed.ac.uk; 5H.E.J. Research Institute of Chemistry, International Centre for Chemical and Biological Sciences (ICCBS), University of Karachi, Karachi 75270, Pakistan; dr.syedabidali@gmail.com; 6Department of Biology, Faculty of Science, Yasouj University, Yasouj 57914, Iran; bfathinia@gmail.com; 7Division of Thrombosis & Hemostasis, Einthoven Laboratory for Vascular and Regenerative Medicine, Leiden University Medical Center, 2333 ZA Leiden, The Netherlands; 8Translational Venomics Group, Madrid Institute for Advanced Studies in Food, E28049 Madrid, Spain; 9Institute for Molecular Bioscience, The University of Queensland, St. Lucia, QLD 4072, Australia

**Keywords:** venom, *Pseudocerastes*, *Eristicophis*, biodiscovery, melanoma, cancer, cytotoxic, haemotoxic, thrombosis, fibrinogenolysis

## Abstract

Venoms are a rich source of potential lead compounds for drug discovery, and descriptive studies of venom form the first phase of the biodiscovery process. In this study, we investigated the pharmacological potential of crude *Pseudocerastes* and *Eristicophis* snake venoms in haematological disorders and cancer treatment. We assessed their antithrombotic potential using fibrinogen thromboelastography, fibrinogen gels with and without protease inhibitors, and colourimetric fibrinolysis assays. These assays indicated that the anticoagulant properties of the venoms are likely induced by the hydrolysis of phospholipids and by selective fibrinogenolysis. Furthermore, while most fibrinogenolysis occurred by the direct activity of snake venom metalloproteases and serine proteases, modest evidence indicated that fibrinogenolytic activity may also be mediated by selective venom phospholipases and an inhibitory venom-derived serine protease. We also found that the *Pseudocerastes* venoms significantly reduced the viability of human melanoma (MM96L) cells by more than 80%, while it had almost no effect on the healthy neonatal foreskin fibroblasts (NFF) as determined by viability assays. The bioactive properties of these venoms suggest that they contain a number of toxins suitable for downstream pharmacological development as candidates for antithrombotic or anticancer agents.

## 1. Introduction

Snake venoms are bioactive secretions that consist mainly of proteinaceous and peptidic toxins, which evolved for use in antagonistic interactions, such as predation and defence [1]. Venom toxin homologues are naturally expressed in body fluids and tissues, and it is thus believed that toxins descend from genes encoding physiological proteins, such as those involved in homeostasis [2]. As a reflection of this ancestry, many venom toxins possess high potency and specificity for their molecular targets [3,4]. This targeted activity constitutes essential criteria in drug design. Accordingly, venoms are rich reservoirs of potential lead compounds. Utilising venoms for biodiscovery is a rapidly growing field with numerous proven successes. For example, enalapril, eptifibatide, tirofiban and batroxobin, all derived from the venoms of vipers, are FDA-approved drugs used for the treatment and/or diagnosis of cardiovascular and haematological disorders [5]. Descriptive studies of venom activity are essential for biodiscovery, yet due to the diversity of venomous snakes and the complexity of their venom, it is estimated that fewer than 0.01% of snake venom compounds have been characterised [6].

The desert vipers of the genera *Pseudocerastes* and *Eristicophis* (Viperidae: Viperinae) are an understudied clade. The limited research devoted to characterising their venoms has described haemolysis and oedema [7], inhibition of platelet aggregation [8,9] and procoagulant activity [10,11] by *E. macmahonii* venom, and anticoagulant and haemorrhagic activity [12], platelet aggregation [13] and cytotoxicity [14] by *P. persicus* venom. Only one study to date has described the venom of *P. urarachnoides*, which is a potent procoagulant by activating both factor X and prothrombin [11]. The activities of these venoms largely adhere to the coagulopathic pathophysiology that is characteristic of viper envenoming. In contrast, presynaptic neurotoxic phospholipase A_2_ proteases dominate the venom of *P. fieldi*, along with absent or very weak coagulopathic activity [12,15,16,17] and anticoagulant properties in some animal plasmas [11]. Furthermore, *P. persicus* and *P. fieldi* venoms have also been found to possess extremely low levels of human factor X and prothrombin activation [11], though their action on human plasma manifests as anticoagulant and the underlying mechanisms are unexplored. Proteomic comparisons corroborated the differential activities among these species’ venoms by revealing extensive variations in their venom protein profiles [11,18].

The wide divergences in venom chemistry within this clade of vipers reflect the underlying evolutionary processes that may have generated biochemical novelty and raises interest for their pharmacological potential. This study sought to explore the venoms for their anticancer properties by assessing their cytotoxicity against the melanoma patient-derived cell line MM96L and healthy skin cell line NFF. In parallel, we investigated their antithrombic potential by incubating the venoms with human plasma and fibrinogen under a range of conditions in colourimetric fibrinolysis assays, thromboelastograms and fibrinogen gels. The venoms showed promise as anticancer or antithrombotic agent candidates, and this study provides the foundation to explore their potential further.

## 2. Results and Discussion

### 2.1. Venom Profile

#### 2.1.1. Proteomics

The protein profiles of the venoms used were previously assessed using 1D SDS-PAGE and mass spectrometry analyses on 2D SDS-PAGE gel spots (Table 1) [11,18]. It was found that, while most toxin groups are conserved among venoms, protein abundances varied considerably. The venoms of *P. urarachnoides* and *E. macmahonii* possess a protein profile typical of vipers, with numerous isoforms of snake venom serine proteases (SVSP), metalloproteases (SVMP), L-amino acid oxidases (LAAO) and lectins. In contrast, *P. fieldi* venom is dominated by phospholipase A_2_s (PLA_2_), while *P. persicus* comprises mostly SVMPs and PLA_2_s. Each venom possesses an abundance of dimers or trimers.

#### 2.1.2. Protease Activity

The venoms varied substantially in their activity on fluorogenic substrates (Figure 1) in corroboration with their proteomic profile [11,18]. The extent of phospholipase and metalloprotease activities is also broadly aligned with the venom content described in prior proteomic studies [11,18]. Assessment of protease activity revealed that serine proteases were the most abundant protease component in *P. urarachnoides* venom while contributing little to the overall protease activity in the other three venoms assayed. This agreed with the proteomic analysis for the *P. urarachnoides* venom, which indicates that serine proteases are featured to a greater extent than in its congeners’ venoms [11]. However, it was incongruent with the extensive expression of serine proteases earlier documented in *E. macmahonii* venom [18]. Further discrepancies between *E. macmahonii* venom proteome and protease activity were evident in its apparent lack of PLA_2_ activity, despite this toxin class having previously been identified in *E. macmahonii* venom [18]. However, vipers commonly express both enzymatic and non-enzymatic isoforms of PLA_2_s. This incongruence observed between assay activity and PLA_2_ abundance may therefore indicate that *E. macmahonii* venom expresses predominantly non-enzymatic PLA_2_ isoforms.

Interestingly, only *P. urarachnoides* demonstrated activity on the SVMP substrate ES001. As this substrate is cleaved by collagenases and gelatinases, ES001 cleavage may represent the degradation of extracellular matrices by haemorrhagic snake venom metalloproteases (SVMPs). However, in vivo haemorrhagic activity has been observed for the venoms of *E. macmahonii* and *P. persicus* [19], despite their absence of ES001 cleavage. This indicates that either the mechanisms by which these species induce haemorrhage vary considerably, or that this substrate is a poor model for haemorrhagic activity. Nevertheless, these assays provided a basic proxy for assessing the relative functional variation of the enzymes constituting these venoms, and thus generally support the variability observed in prior proteomic studies [11,18].

### 2.2. Cytotoxicity

The venoms were tested for their cytotoxic and anticancer properties by incubation with human diseased cells (melanoma cell line MM96L) and healthy cells (neonatal foreskin fibroblasts (NFF)). At concentrations of 5–50 μg/mL, all *Pseudocerastes* venoms significantly reduced the MM96L cell viability and had a minimum effect on NFF cell viability (Figure 2A–C). *Pseudocerastes persicus* venom produced the most striking differential impact on cell viability, in which venom concentrations of 5–50 μg/mL reduced viability of at least 84 ± 2% of MM96L cells while over 85 ± 15% of NFF cells remained viable (Figure 2B). Similarly, 5–50 μg/mL *P. urarachnoides* venom significantly reduced the viability of MM96L cells by at least 80 ± 2%, with over 74 ± 14% NFF cells remaining viable (Figure 2A). Of note, 50 μg/mL of *P. urarachnoides* venom decreased MM96L cell viability by 96 ± 2% yet left 100 ± 0% NFF cells viable. These results demonstrate the potential of *Pseudocerastes* venom compounds to be explored in treatments of melanoma.

Melanoma is an aggressive form of highly metastatic skin cancer with high resistance to chemotherapeutic drugs [20]. We hypothesise that *Pseudocerastes* venoms may target essential signalling cascades for the survival of cells, such as the molecular axis of AKT/PI3K/mTOR. The coordination of these pathways modulates anabolic processes responsible for key components (cholesterol, phospholipids) of the endomembrane system and drives the metabolic process that sustains the generation of new cancer cells. In relation to this, *Pseudocerastes* venoms may also target the integrity and function of the mitochondria, organelles that provide the molecular scaffolds (Acetyl-CoA, amino acids)—and sometimes part of energy sources—necessary for the proliferation of cancer cells.

Similar observations were recently made for an *Octopus Kaurna*-derived peptide, Octpep-1, which specifically targets melanoma cells and with minimum effect on healthy fibroblasts. Octpep-1 exerts its antiproliferative profile by inhibiting the PI3K/AKT/m TOR signalling pathway in melanoma cells [21]. In addition, gomesin (spider) peptides show a similar anti-melanoma pattern with almost no effect on healthy fibroblasts. Specifically, gomesin peptides diminish the viability of melanoma cells by inhibiting the MAPK pathway while simultaneously stimulate the HIPPO pathway [22]. In the current study, the mechanism *Pseudocerastes* venoms utilise to abolish melanoma cells remains unknown. However, the data presented in this study as well as in our previous studies and in the literature warrants further investigations, with special emphasis on MAPK or PI3K/AKT/mTOR pathways, due to their significance in melanoma [23].

A cytotoxic effect specific for human melanoma cells was not observed for *E. macmahonii* venom, which was consistently equally cytotoxic across both cell lines (*p* > 0.01) and consistently reduced viability of at least 70% of cells, even at low concentrations (5 μg/mL) (Figure 2D). This venom contains large quantities of serine proteases and metalloproteases along with smaller quantities of L-amino acid oxidases, each of which are commonly associated with cellular and tissue degradation. However, these toxins are also abundant in the venom of *P. urarachnoides*, which displayed selective cytotoxicity between the two cell lines. This points to key differences within the venom content between the two genera and suggests that the toxins responsible for the variable cytotoxic activity may have originated within the common ancestor of *Pseudocerastes*.

### 2.3. Coagulant Activity

#### 2.3.1. Coagulation and Fibrinolysis

The venoms assayed in the current study have been previously shown to exert different haemotoxic activities in human plasma, such as clot inhibition by *P. fieldi* and *P. persicus* and clot stimulation by *P. urarachnoides* and *E. macmahonii* [11]. Each venom’s effect on clot dynamics was assessed to investigate this further. Fibrinolytic toxin activity (the lysis of fibrin clots as opposed to that of fibrinogen molecules) has been described for the venom of numerous vipers [24,25] and thus was tested by incubating venoms and human plasma with and without human tissue factor, which initiates clot formation, and human tissue plasminogen activator, which initiates clot lysis.

All venoms interfered with the activity of tissue factor and tissue plasminogen activator (Figure 3A), and their haemotoxic properties were largely congruent with those observed in the previous study that analysed these venoms [11]. For example, in the presence of tissue factor, the anticoagulant *P. persicus* and *P. fieldi* venoms prolonged the time to clot formation, with the latter inducing a low-density clot, indicative of a weakened clot structure (Figure 3B). In contrast, the procoagulant *P. urarachnoides* and *E. macmahonii* venoms stimulated high-density clots and rapid fibrin clot formation in the absence of a tissue factor trigger (Figure 3B–D). However, none of the venoms were able to initiate clot lysis, as indicated by the lack of decrease in optical density over time in the absence of tissue plasminogen activator (Figure 3B,D).

#### 2.3.2. Phospholipid Interactions

Interactions between venoms and an anionic phospholipid surface were assessed by conducting a PCPS dilution series and measuring lag time (onset of fibrin clot formation), time to peak (clot formation rate) and peak (clot density maximum) of clotting via turbidity assessment. There were no significant relationships among anionic phospholipid concentration and lag time, time to peak, or clot density in the plasma or thrombin controls (*p* < 0.05 for all relationships) (Figure 4A,B). Increasing phospholipid concentrations between 3 and 100 μM quickened the clotting activity of each *Pseudocerastes* venom. Notably, there was a strong positive relationship between phospholipid concentration and clotting function of *P. fieldi* venom, whereby saturating the venom and plasma with phospholipid vesicles abolished the anticoagulant activity (Figure 4D). A similar (though less pronounced) trend was evident within *P. persicus* test conditions (Figure 4E). This strongly suggests that the dominant anticoagulant mechanism of these venoms in human plasma occurs via antagonistic interactions with phospholipids, which play an essential role in the coagulation cascade [26]. A common anticoagulant mechanism by PLA_2_s is the hydrolysis of plasma phospholipids (e.g., [27]), though non-hydrolytic mechanisms have also been reported (e.g., [26]). Further studies are therefore required to determine the specific interactions occurring by these venoms.

The phospholipid had limited effect upon the activity of the *E. macmahonii* venom except for a significant reduction in clot peak density over increasing phospholipid concentrations, from an optical density (OD) of 0.76 ± 0.02 at 0 μM phospholipid to 0.65 μ 0.01 at 300 μM (non-linear regression: Syx = 0.02893; F-test: F_1,12_ = 12.94, *p* = 0.0037) (Figure 4C). A similar inverse relationship between clot peak and phospholipid concentration was evident within *P. urarachnoides* venom conditions (non-linear regression: Syx = 0.05813; F-test: F_1,22_ = 3662, *p* < 0.0001) (Figure 4F). This could be occurring due to a reduction in fibrin branching or fibrin strand diameter and subsequent fibrin mesh porosity, each of which is known to be affected by relative concentrations of cofactors, activators and the rate of fibrinogen cleavage [28].

These data demonstrate the high variability of venoms’ coagulant activity relative to the chosen model conditions, even within a single plasma. Turbidity measurement of clot density is commonly used in preclinical venom assessments to determine coagulant activity. The variability presented by these viper venoms indicates that such methods should be interpreted with extreme caution by studies relying only on this type of assessment. Such analyses should be run under a range of conditions to determine the consistency of observed clotting response and should ideally be paired with additional clotting assays.

#### 2.3.3. Fibrinogenolytic Activity

Fibrinogen gels and fibrinogen-only thromboelastograms were conducted to evaluate the fibrinogenolytic activity of venoms. Within the thromboelastography, for venoms that did not induce clotting within the first 30 min, thrombin was added to quantify the remaining functional fibrinogen. The thromboelastography indicated that none of the venoms clot fibrinogen directly. Post hoc addition of thrombin revealed that all four venoms have degradative fibrinogenolytic activity, each reducing both clot strength (maximum amplitude—millimetres (mm)) and clot formation rate (angle—degrees) (Figure 5).

*Eristicophis macmahonii* venom possessed the strongest fibrinogenolytic activity, producing a 38 ± 3% reduction in clot strength compared to the control (unpaired t-test: t(6) = 7.901, *p* = 0.000008), followed by *P. urarachnoides* venom (33 ± 6% reduction; unpaired t-test: t(6) = 9.233, *p* = 0.000091). *Pseudocerastes fieldi* and *P. persicus* venoms were comparable for fibrinogenolytic activity. This effect was moderate to low, but significant, with a 22 ± 10% and 18 ± 10% reduction in clot strength, respectively (control vs. *P. fieldi* unpaired t-test: t(6) = 3.838, *p* = 0.008581; control vs. *P. persicus* unpaired t-test: t(6) = 3.244, *p* = 0.017599). These levels of fibrinogenolytic activity were too low to solely account for the anticoagulant effects of the venoms in plasma. Therefore, these modest reductions further support the notion that the clot inhibition by these venoms in human plasma is largely due to PLA_2_ interactions with plasma phospholipids, in conjunction with a fibrinogenolytic activity.

Analysis of fibrinogenolytic activity using SDS-PAGE indicated that *P. persicus* venom possesses the most rapid fibrinogenolytic activity on the fibrinogen Aα-chain, completely degrading this chain within 5 min (Figure 6C). The migration pattern of the fibrinogen degradation products suggested a greater similarity in fibrinogen cleavage sites between *P. urarachnoides* (Figure 6D) and *E. macmahonii* (Figure 6A) venom than between sister species *P. urarachnoides* and *P. persicus*. *Pseudocerastes fieldi* lacked discernible activity in these gels (Figure 6B). The venoms of *P. persicus*, *P. urarachnoides* and *E. macmahonii* all preferentially cleaved the Aα-chain of fibrinogen, followed by the Bβ-chain. This temporal pattern of preferential cleavage follows the dominant cleavage trend amongst vipers. Such fibrinogenases are also known to commonly possess haemorrhagic activity [24]. The fibrinogenolysis observed may thus represent “promiscuous” activity of haemorrhagins, which have been previously described from the venom of *P. persicus* and *E. macmahonii* [19]. Bioactivity testing of venom fractions would be illuminating in this regard.

Within vipers, α-fibrinogenases are typically associated with SVMPs (particularly the P-I class) and β-fibrinogenases with SVSPs [24]. Post hoc fibrinogen gels conducted with inhibitors for serine proteases (PMSF), snake venom metalloproteases (EDTA) and phospholipases (Varespladib) indicated that most of these venoms’ fibrinogenolysis occurs via SVMP activity to the Aα-chain (Figure 7B) with some serine protease activity on the βB-chain (Figure 7A), congruent with the aforementioned trend observed in viperid fibrinogenases [24]. A βB-chain serine protease appeared to be present in the venom of *E. macmahonii*. An interesting result was observed in the activity of the *E. macmahonii* venom with and without PLA_2_ inhibitor Varespladib (Figure 7C), in which inhibition of PLA_2_s appeared to retain a small proportion of the βB-chain. The typical mode of PLA_2_ activity is the hydrolysis of lipoprotein and cell membrane phospholipids [3]. The proteolytic activity of PLA_2_s on lipid-free proteins has been described [29], though it appears to be a rare and poorly understood phenomenon. Whether this finding represents a similar, uncharacteristic PLA_2_ activity is not clear. This would benefit from further investigation as it may represent a currently undescribed mechanism of snake venom PLA_2_s.

Another notable result was observed for *P. fieldi*, with serine protease inhibition eliciting fibrinogenolytic activity on the Aα-chain that was not evident in any of the other conditions assessed. This curious finding could suggest that the venom may contain a serine protease that inhibits a fibrinogenolytic protein, though this would be extremely unusual. However, fibrinogenolysis is evident in the thromboelastography. The discrepancy between the apparent lack of fibrinogenolytic activity without inhibitors (Figure 7B) and the activity observed in the thromboelastography (Figure 5) may ultimately be explained by the putative serine protease involved—though precisely how this may be occurring is puzzling and would benefit from a comprehensive investigation.

## 3. Materials and Methods

### 3.1. Venom Samples

Venom samples from the same snake species were pooled to account for any potential variation in toxin expression between individuals. *Eristicophis macmahonii* venom was collected from three adult male snakes from the Nushki district (30.12° N 67.01° E) of Balochistan, Pakistan. *Pseudocerastes urarachnoides* venoms were collected in Iran under the approval of NIMAD # 942485. *Pseudocerastes fieldi* and *P. persicus* (Pakistan) venoms were purchased from Latoxan (Portes-lès-Valence, France). All venom work was undertaken under the University of Queensland IBSC approval #IBC134BSBS2015. The lyophilised venoms were reconstituted to a working stock concentration of 1 mg/mL in 50% deionised water (diH_2_O) and 50% glycerol (>99.5% purity, Sigma-Aldrich, St. Louis, MO, USA) to reduce enzyme degradation associated with freeze–thaw cycles. These working stock aliquots were stored at −20 °C and used for all subsequent analyses.

### 3.2. Fluorometric Enzyme Activity Assays

Fluorescence assays were performed using a Fluoroskan Ascent™ Microplate Fluorometer with Ascent® Software v2.6 (Thermo Fisher Scientific, Vantaa, Finland). For these assays, venom working stocks were diluted to the desired concentration in dilution buffer: 150 mM NaCl, 50 mM Tris-HCl, pH 7.4, unless otherwise stated.

#### 3.2.1. Snake Venom Phospholipase A_2_ Activity (PLA_2_)

The phospholipase A_2_ (PLA_2_) activity of the venoms was assessed using an EnzChek^®^ Phospholipase A_2_ Assay Kit (Thermo Fisher Scientific, Rochester, NY, USA). Briefly, 10 µL venom (0.01 µg/μL) was brought up to 12.5 µL in PLA_2_ reaction buffer (250 mM Tris-HCL, 500 mM NaCl, 5 mM CaCl_2_, pH 8.9) and loaded in triplicate on a 384-well plate. In each well, 12.5 µL quenched 1 mM EnzChek^®^ Phospholipase A_2_ substrate was added (total volume 25 µL/well), and the run was immediately initiated. Fluorescence was measured for 100 cycles at an excitation of 485 nm and emission of 520 nm at room temperature. A negative control consisted of 12.5 µL PLA_2_ reaction buffer and 10 µL substrate and a positive control of 12.5 µL PLA_2_ reaction buffer and 10 µL kit-provided bee venom.

#### 3.2.2. Snake Venom Metalloprotease (SVMP) Activity

Ten microlitres of venom (0.05 µg/μL) was plated in triplicate on a 384-well plate. Negative or vehicle control consisted of 10 µL dilution buffer. Machine incubation temperature was set at 37 °C. The plate was loaded, and 90 µL quenched substrate was automatically dispensed (10 µL substrate (Fluorogenic Peptide Substrate Cat#ES001; R&D systems, Minneapolis, MN, USA)) in 5 mL enzyme buffer (150 mM NaCl, 50 mM Tris-HCl, 5 mM CaCl_2_, pH 7.4). Fluorescence was monitored at an excitation of 390 nm and emission of 460 nm for 400 cycles or until activity ceased. The assay was repeated using Fluorogenic Peptide Substrate Cat#ES002 (R&D systems, Minneapolis, MN, USA) under identical test conditions.

#### 3.2.3. Snake Venom Serine Protease (SVSP) Activity

Test conditions were identical as those for testing SVMP activity, except that Fluorogenic Peptide Substrate Cat#ES0011 (R&D systems, Minneapolis, MN, USA) was used and fluorescence was monitored at 320 nm (excitation) and 405 nm (emission).

### 3.3. Cytotoxicity

Human neonatal foreskin fibroblast (NFF) and human malignant melanoma (MM96L) cell lines were previously established from patients according to approved ethical procedures and standards of use and compliance by the QIMR Berghofer MRI Human Research Ethics Committee (HREC) under project approval P949. Cells were maintained in RPMI medium supplemented with 1% penicillin–streptomycin and foetal calf serum (FCS), 10% FCS for NFF and 5% FCS for MM96L. Cells were split 24 h prior to the experiment using 0.25% trypsin and seeded in 96-well flat-bottom plates at a density of 5000 and 2500 cells/well for NFF and MM96L cells, respectively. Plates were incubated overnight at 37 °C in a 5% CO_2_-95% humidified environment prior to treatment.

#### MTT Assays

Cell viability was evaluated using colourimetric MTT (Thiazolyl Blue Tetrazolium Bromide; Sigma-Aldrich, St. Louis, MO, USA) assays and as previously described [30]. Briefly, 50 µg/mL, 10 µg/mL, 5 µg/mL, or 1 µg/mL of venom was added to cells (*n* = 4) and incubated for 48 h. Subsequently, MTT (5 mg/mL) was added to each well. Ten percent sodium dodecyl sulphate (SDS) was used as a positive control (100% toxicity), and absorbance was read at 570 nm. Three independent experiments were performed with a minimum of three replicates per treatment. Cell viability readings were normalized against untreated control cells and subtracted from wells containing only media.

### 3.4. Venom-Induced Coagulant Activity

#### 3.4.1. Fibrinolysis

Small unilamellar phospholipid vesicles (PCPS) composed of 75% (*w*/*w*) hen egg L-α-phosphatidylcholine and 25% (*w*/*w*) porcine brain L-α-phosphatidylserine (Avanti Polar Lipids, Alabaster, AL, USA) were prepared and characterized as described [31]. For the fibrinolysis assay, 6 pM tissue factor (TF) (Innovin®, Siemens Healthcare Diagnostics, NY, USA) and 10 μM PCPS were maintained at 37 °C for one hour in HEPES buffer (25 mM HEPES, 137 mM NaCl, 3.5 mM KCl, 0.1% BSA (Bovine Serum Albumin A7030, Sigma Aldrich, St. Louis, MO, USA), pH 7.4). Following this, 17 mM CaCl_2_ and 37.5 U/mL tissue plasminogen activator in dilution buffer were added. To measure the fibrinolytic activity of venoms, 30 μL of the reaction mix, 10 μL of HEPES buffer and 10 μL of venom at three concentrations (0.01 μg/μL, 0.05 mg/μL, 0.1 μg/μL) were loaded in triplicate into a 96-well plate. Control wells were loaded following the same experimental protocol, with the substitution of tissue factor and/or tissue plasminogen activator by equal volumes of HEPES buffer. Fifty microlitres of normal human platelet-poor pooled plasma (3.2% (*w*/*v*) sodium citrate) (Sanquin, Amsterdam, The Netherlands) was loaded into each well, and the clot formation and subsequent lysis were monitored by measuring the optical density at 405 nm every 30 s for 3 h at 37 °C on a Spectra Max M2e microplate reader using Softmax Pro software (Molecular Devices, Sunnyvale, CA, USA).

#### 3.4.2. Phospholipid Interactions

Interactions between venom and anionic phospholipid vesicles on fibrin clot formation were conducted as per fibrinolysis assays, but without the addition of tissue plasminogen activator to venom wells and with an extended concentration range of PCPS across wells (serial dilution: 0, 1, 3, 10, 30, 100 and 300 μM). Reactions in which venom was replaced by thrombin (0.01 μg/μL) were used as a positive control. The lag time was defined as the time point at which the optical density at 405 nm increased (delta absorbance >0.04); the peak time was the time at maximum optical density.

#### 3.4.3. Fibrinogenolysis

One millimolar 12% SDS-PAGE gels were prepared using pre-established protocols and reagents described in op den Brouw et al. [11]. Human fibrinogen was reconstituted to a concentration of 2 mg/mL in isotonic saline solution, flash-frozen in liquid nitrogen and stored at −80 °C until use. The following assay was conducted in triplicate for each venom: five aliquots containing 10 μL reducing buffer (5 μL of 4× Laemmli sample buffer (Bio-Rad, Hercules, CA, USA), 5 μL diH_2_O, 100 mM DTT (Sigma-Aldrich, St. Louis, MO, USA)) were prepared. Then, a stock aliquot of 50 μL fibrinogen (1 mg/mL in buffer: 150mM NaCl, 50 mM TrisHCl, 5mM CaCl, pH 7.4) was warmed to 37 °C in an incubator. Once warmed, 10 μL was removed from the stock aliquot and added to one of the prepared aliquots, manually mixed and boiled at 100 °C for 4 min to represent the “0 min incubation” fibrinogen control. Four microlitres of venom (1 mg/mL) was added to the stock aliquot of fibrinogen (amounting to approximately 0.1 mg/mL of venom and 1 mg/mL of fibrinogen in 40 μL total volume), mixed and immediately returned to the incubator for the duration of the assay. At each incubation time period (1, 5, 20 and 60 min), 10 μL was taken from the stock aliquot, added to an aliquot containing 10 μL reducing buffer, pipette mixed and boiled at 100 °C for 4 min. These aliquots were then loaded into the gels and run using 1× gel running buffer at room temperature for 20 min at 90 V (Mini Protean3 power-pack from Bio-Rad, Hercules, CA, USA) and then 120 V until the dye front approached the bottom of the gel. Gels were stained with colloidal coomassie brilliant blue G250 (34% methanol (VWR Chemicals, Tingalpa, QLD, Australia), 3% orthophosphoric acid (Merck, Darmstadt, Germany), 170 g/L ammonium sulphate (Bio-Rad, Hercules, CA, USA), 1 g/L coomassie blue G250 (Bio-Rad, Hercules, CA, USA) and de-stained in diH_2_O.

Fibrinogen gels were run with and without the SVMP inhibitor Ethylenediaminetetraacetic acid (EDTA) (Sigma Aldrich, St. Louis, MO, USA) (10 mM), the serine protease inhibitor phenylmethylsulphonyl fluoride (PMSF) (Sigma Aldrich, St. Louis, MO, USA) (1 mM in isopropanol) and the PLA_2_ inhibitor Varespladib (Sigma-Aldrich, St. Louis, MO, USA) (10 mM in DMSO) to investigate the comparative fibrinogenolytic activity of SVMP, SVSP and PLA_2_ toxin classes. Testing conditions were identical to those for fibrinogen gels without inhibitors, except 5 μL venom (1 mg/mL) or 5 μL buffer (negative control) was added to 5 μL fibrinogen (2 mg/mL) with/without 1 μL inhibitor and incubated for 20 min only. This increase in venom concentration and decrease in total incubation time was to account for the rapid half-life (15–60 min) of PMSF.

The effects of the venoms on the fibrinogen were quantified using a Thromboelastograph^®^ 5000 Haemostasis analyser (Haemonetics^®^, Haemonetics Australia Pty Ltd., Sydney, Australia). Human fibrinogen (Sigma Aldrich, St. Louis, MO, USA) was reconstituted to a concentration of 4 mg/mL in isotonic saline solution (150mM NaCl, 50mM Tri-HCl (pH 7.3)), flash-frozen in liquid nitrogen and stored at −80 °C until use. In quadruplicate, 189 μL of fibrinogen was combined with 72 μL CaCl_2_ (25 mM stock solution Stago Cat# 00367 STA, Asniéres sur Seine, France), 72 μL phospholipid (STA C·K Prest standard kit, Stago Cat# 00597, Asniéres sur Seine, France; solubilised in Owren Koller (OK) buffer (Stago, Asniéres sur Seine, France)), and 20 μL OK buffer in an assay cup (Haemonetics Australia Pty Ltd., North Rye, Sydney, Australia). Then 7 μL of venom working stock (1 mg/mL) was added to the cup, pipette mixed and the machine was immediately started. Tests were run at 37 °C and measured for 30 min. If no clots were formed by the venoms, 7 μL thrombin (STA^®^-Liquid Fib, Stago Cat#00673, Asniéres sur Seine, France) was added post hoc and the measurements were continued for another 30 min to assess fibrinogen degradation. Negative controls followed identical methods, except 7 μL of 1:1 glycerol:diH_2_O replaced venom.

### 3.5. Statistics

Data were graphed and analysed using GraphPad Prism version 8.0.0 for iOS (GraphPad Software, San Diego, CA, USA). For phospholipid interaction data, curves were fit using the non-linear regression (linear model or log(agonist) vs. response (three parameters)) with confidence intervals set to 95% and Shapiro–Wilk, Kolmogorov Smirnov and Q-Q plot diagnostic tests for normality. For linear models, extra sum-of-squares F-tests (significance set to *p* < 0.05) were run to compare slope deviation from 0. Venom-fibrinogen thromboelastography data were tested for normality using Shapiro–Wilk tests and compared to the control using unpaired t-tests. For graphing, the venom data were then normalised to the thrombin control (100%) and subtracted from 100 to calculate percentage reduction in functional fibrinogen. For the MTT assays, venom cytotoxicity data were normalised to the vehicle-treated cells negative control (100%) to calculate the percentage of viable cells remaining following venom treatment. Data were then tested for normality using Shapiro–Wilk tests (significance set to *p* < 0.05) and Q-Q plot diagnostic tests. The percentages of viable cells were then compared to the control using unpaired t-tests.

## 4. Conclusions

This study revealed promising avenues for further research into the pharmacological potential of compounds from these venoms. Each of the *Pseudocerastes* venoms significantly reduced the viability of the cancerous MM96L cells while having a limited effect on healthy NFF cells, showing specificity of these venoms towards tumorous cells. This was particularly pronounced in the venoms of *P. persicus* and *P. urarachnoides*, indicating that they possess a toxin that shows therapeutic potential for melanoma. Future studies should be aimed at identifying the toxins and molecular mechanisms underpinning the selective cytotoxicity of the crude *Pseudocerastes* venoms in addition to tests on a wider range of melanoma cell lines of different mutations.

The anticoagulant activities of *P. fieldi* and *P. persicus* venoms appeared to be predominately driven by interactions between PLA_2_s and plasma phospholipids with a limited degree of fibrinogenolysis. Fibrinolysis was not evident by these venoms. However, variable actions upon fibrinogen were present, with the possibility of these being novel mechanisms. Modest evidence indicated that some fibrinogenolytic activity may be mediated by *E. macmahonii* venom phospholipases, which represents an atypical activity for this toxin class and warrants further investigation. Furthermore, the results suggested the possibility of a venom-derived serine protease that inhibits fibrinogenolytic activity in the venom of *P. fieldi*. The use of drugs in the treatment of clotting disorders and the management of bleeding during surgical procedures can be complicated by the absence of an adequate inhibitor or reversal agent. An optimal treatment could consist of an active compound paired with a highly specific inhibitor. The fibrinogenolytic toxin and associated serine protease inhibitor that may be present in this venom could represent the foundations of such a combinatory treatment/reversal strategy.

While future studies are required to characterise key venom components comprehensively, the bioactive properties of the crude venoms used in this study suggest that they contain a number of toxins that may be suitable for pharmacological development into antithrombotic or anticancer agents.

## Figures and Tables

**Figure 1 ijms-22-06896-f001:**
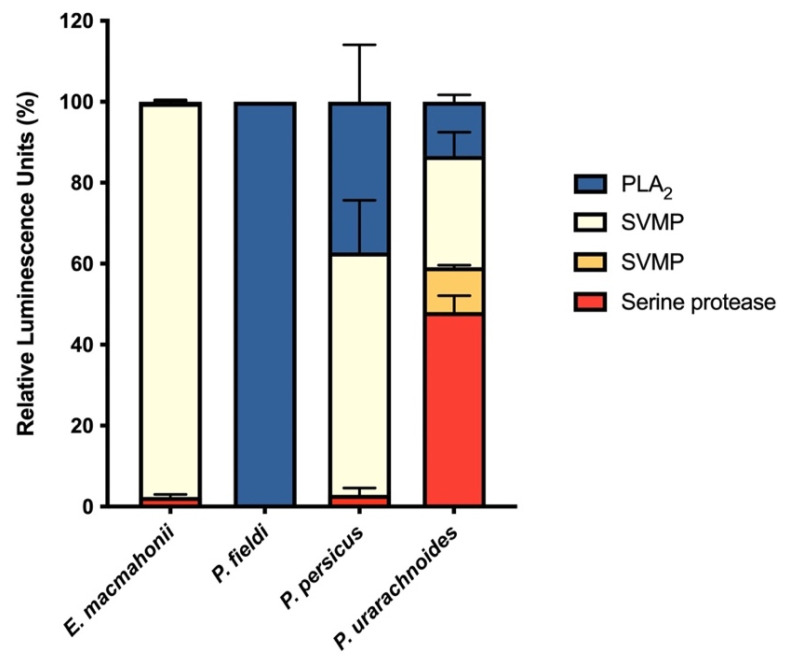
Relative protease activity of *Pseudocerastes* and *Eristicophis* venoms on four synthetic substrates for non-comprehensive assessment of snake venom metalloprotease (SVMP) using substrate ES001 (light yellow) or ES002 (dark yellow) (crude venom: 0.5 ng/mL), serine protease using substrate ES011 (red) (crude venom: 0.5 ng/mL), or phospholipase (PLA_2_) (blue) (crude venom: 4 ng/mL) activities. The data are presented as relative luminescence units (%) normalised to the total luminescence detected for all four protease assays combined and are the mean ± SD of four independent experiments.

**Figure 2 ijms-22-06896-f002:**
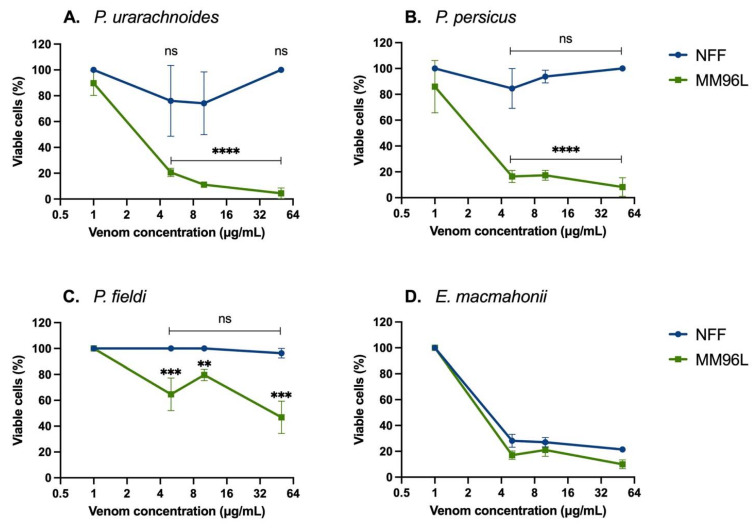
Viable human cells (%) after incubation with *Pseudocerastes* (**A–C**) and *Eristicophis* (**D**) venoms (1, 5, 10, or 50 µg/mL) for 48 h, quantified by MTT assays. Cell viability readings were normalised against vehicle-treated control cells (data not shown). Venom-treated cell data were tested for normality using Shapiro–Wilk tests, then compared to their controls using unpaired t-tests as they were normally distributed. The data are expressed as mean ± SD and are the result of at least three replicates. **** = *p*  <  0.0001, *** = *p* < 0.001, ** = *p* < 0.01, ns = no significant difference between venom-treated (data shown) vs. vehicle-treated control (data not shown) cells. MM96L = melanoma cells, NFF = neonatal foreskin fibroblasts.

**Figure 3 ijms-22-06896-f003:**
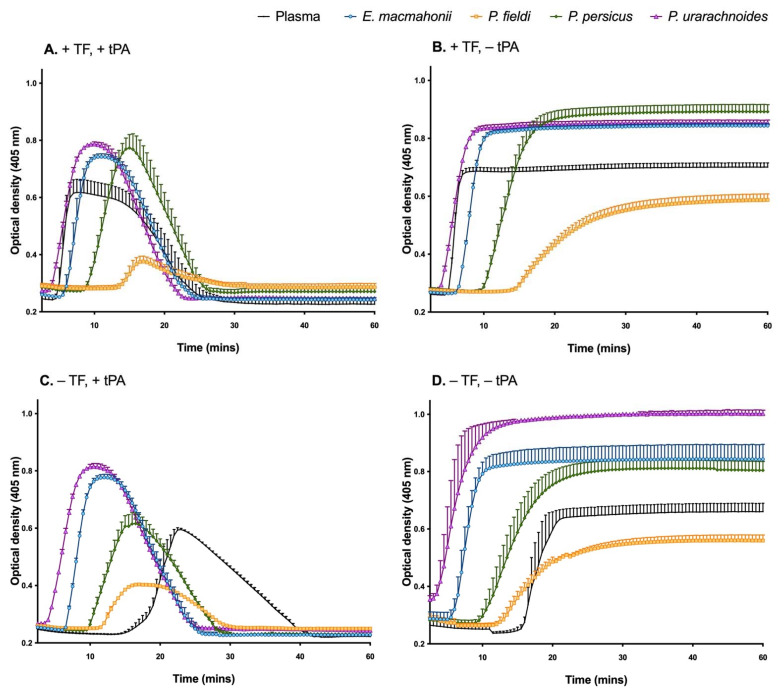
Plasma clot formation and fibrinolysis of human plasma were followed by turbidity assessment at 405 nm in the presence of 0.1 μg/mL *Pseudocerastes* or *Eristicophis* venom with (**A**,**B**) or without (**C**,**D**) tissue factor (TF) and with (**A**,**C**) or without (**B**,**D**) tissue plasminogen activator (tPA), measured over 60 min at 37 °C. Plasma without venom was used as a control. The data are the mean ± SD of three independent experiments.

**Figure 4 ijms-22-06896-f004:**
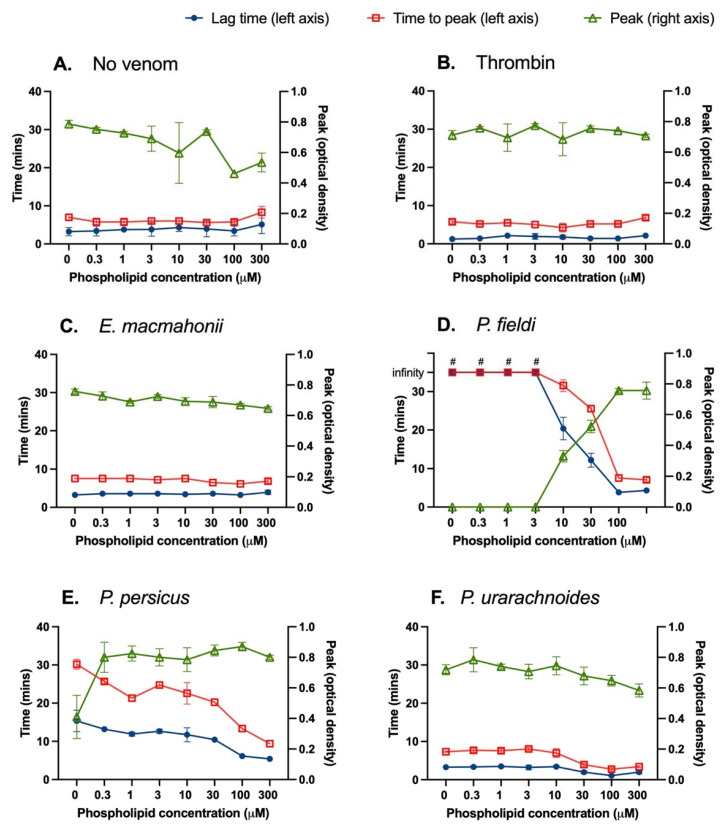
Plasma clot formation of human plasma was followed by turbidity assessment at 405 nm in the presence of tissue factor (6 pM) in the absence or presence of 0.01 μg/μL of *Eristicophis* (**C**), *Pseudocerastes* (**D**–**F**) venoms were measured for 60 min at 37 °C. The lag time (blue circles, left axis), time to peak (red squares, left axis) and peak (optical density at 405 nm; green triangles, right axis) are shown. Reactions in which venom was replaced by thrombin (0.01 μg/μL) (**B**) or vehicle solution (**A**) were used as positive and negative controls, respectively. The data are the mean ± SD of three independent experiments. # = infinity, i.e., no clot occurred during the 35 min measurement.

**Figure 5 ijms-22-06896-f005:**
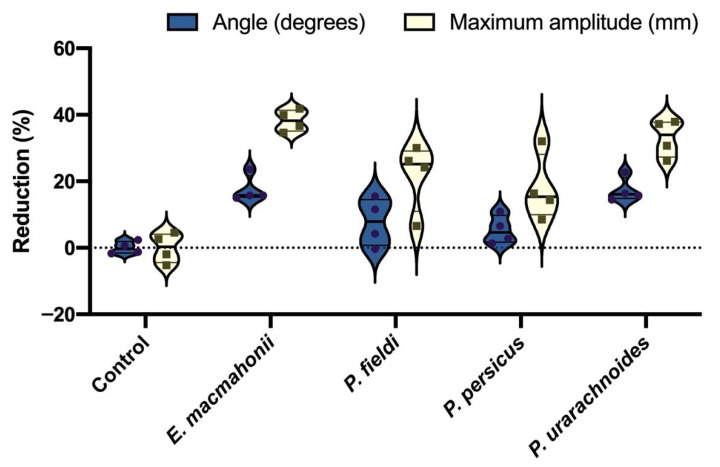
Reduction (%) of functional human fibrinogen (2 mg/mL) following 30 min of incubation with venom (20 μg/mL), assessed by post hoc addition of thrombin with subsequent clotting measured by thromboelastography. Angle represents the reduction in the rate of clot formation upon addition of thrombin; maximum amplitude represents the reduction in clot strength. Control = fibrinogen incubated for 30 min without venom prior to addition of thrombin. The data represent four individual analyses per condition; violin plot shows distribution of data.

**Figure 6 ijms-22-06896-f006:**
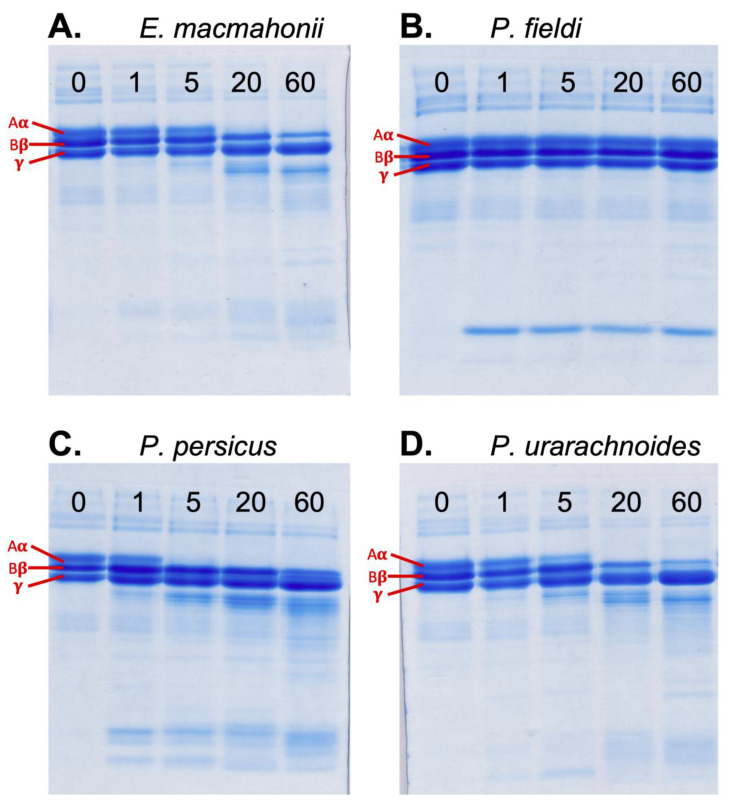
Activity of *Eristicophis* (**A**) and *Pseudocerastes* (**B**–**D**) venoms on human fibrinogen (0.1 μg/mL venom in 1 mg/mL fibrinogen) after 1, 5, 20 and 60 min of incubation at 37 ºC. The fibrinogen control (no venom) is shown at time point 0. Aα = Aα (A-alpha) chain of fibrinogen, Bβ = Bβ (B-beta) chain of fibrinogen, ϒ = ϒ (gamma) chain of fibrinogen. The panels depict one of the three replicates conducted with each venom.

**Figure 7 ijms-22-06896-f007:**
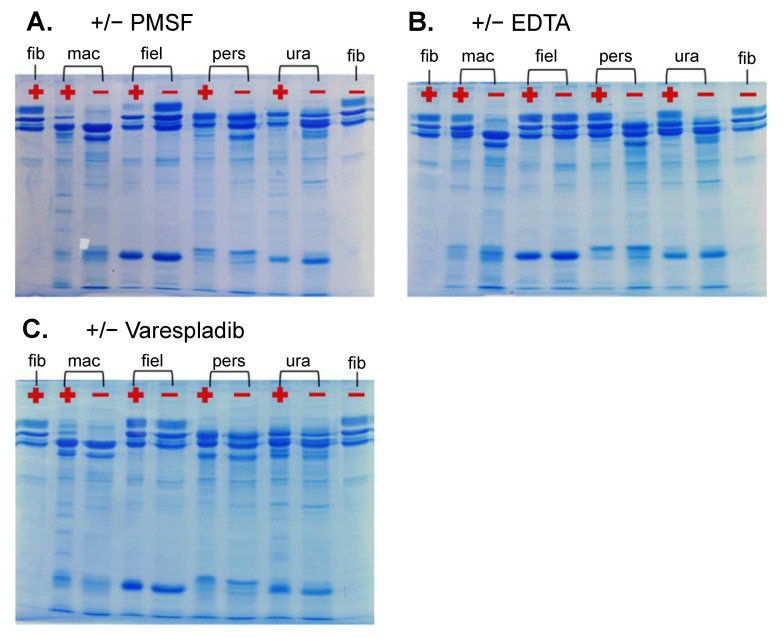
Activity of *Eristicophis* and *Pseudocerastes* venoms on human fibrinogen (0.5 μg/mL venom in 1 mg/mL fibrinogen) after 20 min of incubation at 37 °C, with (+) and without (–) inhibitors for (**A**) serine proteases (1 mM PMSF); (**B**) metalloproteases (10 mM EDTA) and (**C**) phospholipases (10 mM Varespladib). fib = fibrinogen control (no venom), mac = *E. macmahonii*, fiel = *P. fieldi*, pers = *P. persicus*, ura = *P. urarachnoides*.

**Table 1 ijms-22-06896-t001:** Toxin classes identified following mass spectrometric analyses of desert viper 2D SDS-PAGE venom profiles [11,18].

Toxin Type	*E. macmahonii*	*P. fieldi*	*P. persicus*	*P. urarachnoides*
CRiSP	X	X	X	n.d.
Disintegrin	X	n.d.	n.d.	n.d.
SVSP	X	X	X	X
Kunitz peptide	n.d.	X	X	n.d.
LAAO	X	n.d.	n.d.	X
Lectin	X	X	X	X
Natriuretic peptides	n.d.	n.d.	n.d.	n.d.
NGF	n.d.	X	X	n.d.
Nucleotidase	n.d.	X	n.d.	n.d.
PLA_2_	X	X	X	X
SVMP	X	n.d.	X	X
VEGF	X	X	X	n.d.

CRiSP = Cysteine-rich secretory protein; SVSP = Snake venom serine protease; LAAO = L-Amino acid oxidase; NGF = Nerve growth factor; PLA_2_ = Phospholipase A_2_; SVMP = Snake venom metalloprotease; VEGF = Vascular endothelial growth factor; n.d. not detected.

## Data Availability

The data presented in this study are available upon request to the corresponding authors.

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
