# Peer review of "Pharmacological Characterisation of Pseudocerastes and Eristicophis Viper Venoms Reveal Anticancer (Melanoma) Properties and a Potentially Novel Mode of Fibrinogenolysis"

_ijms, 2021, doi:10.3390/ijms22136896_

Round 1

Reviewer 1 Report

The authors of the manuscript focused on antithrombotic potential of snake venoms in hematological disorders and cancer treatment. The authors have come up with interesting results that are very important for medical research.

Page 2, lines 161-168: Bothrops atrox is a highly venomous pit viper species. The reptilase time is a functional clotting assay based upon the enzymatic activity of batroxobin a venom isolated from the this snake. Reptilase time is a key time in the diagnosis of congenital dysfibrinogenemia. Recently, a study was published in patients with dysfibrinogenemia in which reptilase time was used in the diagnosis. In addition to this section, it is important to state in the introduction that snake venoms are used in the diagnosis of congenital fibrinogen disorders. Authors should cite this study: ,, Simurda et al. Comparison of clinical phenotype with genetic and laboratory results in 31 patients with congenital dysfibrinogenemia in northern Slovakia. Int J Hematol. 2020 Jun;111(6):795-802. doi: 10.1007/s12185-020-02842-9.“

Figures and tables in the text are very clearly written.

I have to say that with these 26 references there are only 6 references newer than 5 years old. It is also appropriate to add newer references.

Author Response

We thank the Editor and the Reviewers, who have assessed our manuscript and provided us with their comments. We have now addressed their concerns in a point-by-point response (please see below) and as track changes in the revised manuscript.

We look forward to receiving the final decision regarding the potential publication of our work in the International Journal of Molecular Sciences.

Best regards,

Dr. Bianca op den Brouw

Reviewer 1

The authors of the manuscript focused on antithrombotic potential of snake venoms in hematological disorders and cancer treatment. The authors have come up with interesting results that are very important for medical research.

  1. Page 2, lines 161-168: Bothrops atrox is a highly venomous pit viper species. The reptilase time is a functional clotting assay based upon the enzymatic activity of batroxobin a venom isolated from the this snake. Reptilase time is a key time in the diagnosis of congenital dysfibrinogenemia. Recently, a study was published in patients with dysfibrinogenemia in which reptilase time was used in the diagnosis. In addition to this section, it is important to state in the introduction that snake venoms are used in the diagnosis of congenital fibrinogen disorders. Authors should cite this study: ,, Simurda et al. Comparison of clinical phenotype with genetic and laboratory results in 31 patients with congenital dysfibrinogenemia in northern Slovakia.Int J Hematol. 2020 Jun;111(6):795-802. doi: 10.1007/s12185-020-02842-9.“

We thank the reviewer for providing us with this reference and the helpful background information. We have included this sentence in our revised manuscript to include that drugs derived from venom-toxins are also used as diagnostic tools for disorders of haemostasis, and we have used the suggested reference as an example. We have refrained from further discussing the details of venom-derived drugs and their applications as there are many examples to draw on and we feel that the specifics are superfluous to our predominantly descriptive study.

  1. Figures and tables in the text are very clearly written.

Thank you for the positive feedback.

  1. I have to say that with these 26 references there are only 6 references newer than 5 years old. It is also appropriate to add newer references.

We agree with the reviewer that more recent references would be desirable. However, most of our cited references are about the venoms of snakes used in this study. There is a very small amount of literature on these species and the majority was conducted many years ago. Hence, the reference list contains dated papers.

Furthermore, as our paper is broadly descriptive, we have chosen to cite core reviews and original papers for relevant proteins, mechanisms, or methods. Content that would require consolidation of current research, such as speculative or detailed discussion on specific clinical applications or mechanisms of these venoms, is generally beyond the scope of this fundamentally descriptive study.

We have, however, added five additional references (two in the introduction and three in cytotoxicity section 2.1) that have been published between 2018-2021.

Reviewer 2 Report

In the present study, the authors investigated the anticoagulant and anticancer potential of crude Pseudocerastes and Eristicophis snake venom. The present study showed that the venoms showed the anticoagulant properties presumably through the hydrolysis of phospholipids and by selective fibrinogenolysis. In addition, the authors showed that some of venoms specifically inhibited the growth of melanoma cells. I think that the topic and obtained results of this manuscript is interesting. In my opinion, this manuscript is suitable for the publication in International Journal of Molecular Sciences after minor revision.

Comments:

  1. The description about source of cells used in this study is missing.
  2. 2: The statistical analysis should be performed in not only Fig. 2A&B but also Fig. 2 C&D. In addition, the unit of x-axis should be revised as “μg/mL”.
  3. 2A-C: How do the venoms specifically decrease the growth of melanoma cells? Please discuss the potential mechanisms of it.

Author Response

We thank the Editor and the Reviewers, who have assessed our manuscript and provided us with their comments. We have now addressed their concerns in a point-by-point response (please see below) and as track changes in the revised manuscript.

We look forward to receiving the final decision regarding the potential publication of our work in the International Journal of Molecular Sciences.

Best regards,

Dr. Bianca op den Brouw

Reviewer 2

In the present study, the authors investigated the anticoagulant and anticancer potential of crude Pseudocerastes and Eristicophis snake venom. The present study showed that the venoms showed the anticoagulant properties presumably through the hydrolysis of phospholipids and by selective fibrinogenolysis. In addition, the authors showed that some of venoms specifically inhibited the growth of melanoma cells. I think that the topic and obtained results of this manuscript is interesting. In my opinion, this manuscript is suitable for the publication in International Journal of Molecular Sciences after minor revision.

Comments:

  1. The description about source of cells used in this study is missing.

This information is now included in the text (lines 408-410) and is as follows:

MM96L and NFF cell lines were previously established from patients according to approved ethical procedures and standards of use and compliance by the QIMR Berghofer MRI Human Research Ethics Committee (HREC) under project approval P949.

2. The statistical analysis should be performed in not only Fig. 2A&B but also Fig. 2 C&D. In addition, the unit of x-axis should be revised as “μg/mL”.

We thank the reviewer for their attention to detail. We have amended the x-axis label.

While we did perform statistics on all venoms, we had not included all the results in the figure. We have now included the statistical information in the graph of Fig 2C. However, we have refrained from adding stats to Fig 2D as we would prefer to draw the reader’s attention to the notable promising discoveries in the data.

Furthermore, we feel that adding statistical results to Fig 2D could cause confusion. The statistics presented in Fig 2A-C (ie ‘ns’ or P values) denote the comparisons between venom-treated cells to the controls (as described in the figure caption). For Figure 2D, the venom is strongly cytotoxic on both cell lines, with venom vs control comparisons producing P < 0.0001 for each concentration and cell line.

For the data in Fig 2D, the comparison of importance is that there is no significant difference in cytotoxicity between NFF vs MM96L cells, indicating that this venom has limited potential for cancer treatment. However, it is not possible to include this on the graph, as in this instance “ns” would denote the results of a different comparison to that which is described in the figure caption (i.e., venom-treated NFF cells vs venom-treated MM96L cells for Fig 2D data, as opposed to venom-treated cells vs controls for Fig 2A-C data).

We have, however, added details to the accompanying text for Fig 2D (line 175-176) to indicate that statistics were performed on this venom too.

3. 2A-C: How do the venoms specifically decrease the growth of melanoma cells? Please discuss the potential mechanisms of it.

We thank the reviewer for this suggestion. We have now briefly addressed possible mechanisms of the observed antiproliferative capacity of the snake venoms in melanoma cells in lines 148-173. Our answer is included also here for the reviewer’s perusal.

“We observed a preference of Pseudocerastes venoms to target melanoma cells with minimum effect upon healthy fibroblasts. For this, we believe that Pseudocerastes venoms may target essential signalling cascades for the survival of cells like the molecular axis of AKT/PI3K/mTOR. The coordination of these pathways modulates anabolic processes responsible of key components (cholesterol, phospholipids) of the endomembrane system and drive the metabolic process that sustain the generation of new cancer cells. In relation to this, Pseudocerastes venoms may also target the integrity and function of the mitochondria, organelles that provide the molecular scaffolds (Acetyl-CoA, amino acids) – and sometimes part of energy sources – necessary for the proliferation of cancer cells.

Similar observations were recently made for an Octopus Kaurna-derived peptide, Octpep-1, which specifically targets melanoma cells and with minimum effect on healthy fibroblasts. Octpep-1 exerts its antiproliferative profile by inhibiting the PI3K/AKT/m TOR signalling pathway in melanoma cells (Moral-Sanz et al., 2021). In addition, gomesin (spider) peptides show a similar anti-melanoma pattern with almost no effect on healthy fibroblasts. Specifically, gomesin peptides diminish the viability of melanoma cells by inhibiting the MAPK pathway while simultaneously stimulate the HIPPO pathway (Ikonomopoulou et al., 2018). Hence, in the current study, the mechanism Pseudocerastes venoms utilise to abolish melanoma cells remains unknown. However, the data presented in this study as well as our previous studies and literature warrants further investigations, with special emphasis on MAPK or PI3K/AKT/mTOR pathways, due to their significance in melanoma (Lassen et al, 2014). “

Round 2

Reviewer 1 Report

The presented manuscript has been corrected in response to the suggestions. The authors have followed the recommendations of the reviewer. After the revision, the provided data and addition of the results became more clear. I would like to thank the authors for resubmitting the manuscript and explaining the obscure points from the previous version.